# Application of Heuristic Algorithms in the Tomography Problem for Pre-Mining Anomaly Detection in Coal Seams

**DOI:** 10.3390/s22197297

**Published:** 2022-09-26

**Authors:** Rafał Brociek, Mariusz Pleszczyński, Adam Zielonka, Agata Wajda, Salvatore Coco, Grazia Lo Sciuto, Christian Napoli

**Affiliations:** 1Department of Mathematics Applications and Methods for Artificial Intelligence, Faculty of Applied Mathematics, Silesian University of Technology, 44-100 Gliwice, Poland; 2Institute of Energy and Fuel Processing Technology, 41-803 Zabrze, Poland; 3Department of Electrical, Electronics and Informatics Engineering, University of Catania, Viale Andrea Doria, 6, 95125 Catania, Italy; 4Department of Mechatronics, Silesian University of Technology, 44-100 Gliwice, Poland; 5Department of Computer, Control, and Management Engineering, Sapienza University of Rome, Via Ariosto 25, 00185 Roma, Italy

**Keywords:** computed tomography, inverse problem, optimization, incomplete data set

## Abstract

The paper presents research on a specific approach to the issue of computed tomography with an incomplete data set. The case of incomplete information is quite common, for example when examining objects of large size or difficult to access. Algorithms devoted to this type of problems can be used to detect anomalies in coal seams that pose a threat to the life of miners. The most dangerous example of such an anomaly may be a compressed gas tank, which expands rapidly during exploitation, at the same time ejecting rock fragments, which are a real threat to the working crew. The approach presented in the paper is an improvement of the previous idea, in which the detected objects were represented by sequences of points. These points represent rectangles, which were characterized by sequences of their parameters. This time, instead of sequences in the representation, there are sets of objects, which allow for the elimination of duplicates. As a result, the reconstruction is faster. The algorithm presented in the paper solves the inverse problem of finding the minimum of the objective function. Heuristic algorithms are suitable for solving this type of tasks. The following heuristic algorithms are described, tested and compared: Aquila Optimizer (AQ), Firefly Algorithm (FA), Whale Optimization Algorithm (WOA), Butterfly Optimization Algorithm (BOA) and Dynamic Butterfly Optimization Algorithm (DBOA). The research showed that the best algorithm for this type of problem turned out to be DBOA.

## 1. Introduction

Application of computed tomography can take place wherever there is a need to examine the interior of a certain object without disturbing its structure. Medicine is a classic example of using the computed tomography. In this field, the developed algorithms are so reliable and efficient that reconstruction of the interior structure of the examined patient in sufficient resolution in a short time is not a big challenge for modern computer tomographs. A large part of the papers published in non-medical journals concerns the modification of algorithms in such a way that reducing the dose of radiation received by the patient does not deteriorate the quality of reproduction [1].

In this paper, we deal with a non-medical problem related to the issue of an incomplete data set. Such phenomena are quite frequent, and an example of it may be the study of coal seams. The natural process of coal seams formation may result in the formation of undesirable, for economic reasons, excesses of other rocks or compressed gas tanks hazardous to the health and life of the working crew. The current energy crisis may put pressure on mines to increase production. Higher production, in turn, may be related to the exploitation of previously unexploited deposits. This, however, is associated with the risk of hitting and unsealing such a tank. The uncontrolled gas outburst next to fires, rock bursts and gas explosions is the most common cause of dangerous incidents in mines (in the modern history of Polish hard coal mines, there were several such large incidents, and, in 1979, in one of the Polish mines, almost 20 miners died due to uncontrolled gas outburst). The proposed method of pre-exploitation coal seam examination has two greatest advantages: it can detect dangerous anomalies in the seam, and such examination does not interfere with the seam exploitation (the examination of further parts of the seam can be performed during the operation of the already examined part).

Computed tomography algorithms, apart from the classic medical applications [2] and the examination of coal decks [3,4] mentioned before, have a number of other applications, almost classic, such as X-raying luggage, examining the construction of building materials, or less obvious, such as examining flames, the earth’s crust, seas and oceans, objects in motion or crash tests [5,6,7,8]. The authors studied the algorithms of computed tomography focus on its non-standard approach to an incomplete data set, which occurs, for example, in the case of coal seam research. The conducted research has shown that, despite much poorer information about the interior of the tested object, with the appropriate approach to the problem, the algorithms are convergent and stable, and thus they can be successfully used in the problem of an incomplete data set [9]. The only problem with the algorithms developed in this way is the high computation time, while in classical problems a dozen or so iterations were enough (details are presented in Section 3); in the case of an incomplete data set, this number may increase to several hundred. The authors took steps to prevent this undesirable feature by using, inter alia, chaotic and block algorithms, parallel computing or heuristic algorithms [3,4].

In the classic problem of computed tomography, e.g., in medical tomography, the assumptions of Kotelnikov’s theorem are easily fulfilled. This means that enough X-rays can be taken from sufficiently many angles. However, there are cases when such projections cannot be made. Such a situation may take place, for example, when the tested object is unavailable due to its location (ocean and space research), size (large building structures) or lack of access. A good example of the latter case is the coal bed. The tests on such a coal seam must be performed for economic reasons (then, the presence of undesirable scale in the coal is checked) or for safety reasons. In the latter case, compressed gas is sought in the coal seams, which can rapidly expand during extraction, ejecting rock fragments along with the gas. This is a threat to the health and life of the mining crew. Currently, most of the work to check the presence of such reservoirs of gas is carried out by drilling, but this method is time-consuming and involves significant costs.

If the coal seam examination was performed using computed tomography methods, then the works related to the extraction of coal and the works related to the examination of the coal seam would be carried out in parallel and would not interfere with each other. To prepare a mining wall, auxiliary structures are made for a given seam, on the walls of which a system of sensors sending a beam of penetrating radiation can be placed on the one side, and sensors (receivers) of this signal are placed on the other side (see Figure 1 and Figure 2). On the one hand, such an approach is economically beneficial, but on the other hand, such projections differ significantly from the assumptions for the quality of projection in computed tomography.

Due to the unavailability of the coal seam, the obtained projections are “one-sided”, thus they carry much less information about the examined object. The research carried out so far shows that in such a case “quality” can be made up of with quantity. However, this approach has a disadvantage; many more calculations are needed, and thus the time needed to obtain a satisfactory accuracy is significantly extended. A series of work devoted to eliminate this drawback has brought results, but it seems that the efficiency of computed tomography reconstructive algorithms in the issue of incomplete data set can be significantly increased, which is the goal of this study.

In this paper, we focus on improving one of these approaches [4], where the detected objects are represented by sequences of rectangles characterized by sequences of their parameters. This time, instead of sequences in the representation, there are sets, which allow for the elimination of duplicates of such representations. The considered algorithms were adapted to the task in such a way that they do not operate on the argument sequence, but on sets of fives, where the first pair in the five represents the lower left corner of the rectangle, the second pair—the upper right corner, and the fifth argument is the desired density. As a result, the reconstruction is faster. Additionally, other heuristic algorithms, such as Aquila Optimizer (AO), Firefly Algorithm (FA), Whale Optimization Algorithm (WOA) and Dynamic Butterfly Optimization Algorithm (DBOA) are tested for usability.

## 2. Ideas of Computed Tomography

Computed tomography can be used where there is a need for non-invasive examination of the interior of different objects. Such examination consists of X-raying (this is the most common, but not the only method) of the examined object with penetrating radiation and analyzing the energy loss after passing such radiation through the examined object. Such an analysis can be useful due to the fact that each type of matter has its own individual (different than for other types of matter) coefficient (capacity) of absorbing the energy of penetrating radiation. The measure of this coefficient is often given in Hounsfield units [HU]. Examples of the values of this coefficient for X-ray radiation can be found in [3].

The mathematical description of the measure of such a loss for one radius *L* (projection) is given by the formula:(1)$$p_{L}=\ln\frac{I_{0}}{I}\int_{L}f\left(x,y\right)\;dL,$$
where $p_{L}$ is the projection obtained on the path *L* of the given ray, $I_{0}$ is the initial radiation intensity, *I* is the final intensity (after passing along the path *L* of the ray through the tested object), and the function *f* is the density distribution of the tested object. Obviously, the information obtained from one ray does not allow for reconstructing the interior of the examined object along the path of this ray. Such X-rays should be done sufficiently, not only in terms of quantity, but also in terms of quality understood as the number of X-rays angles. The first studies of computed tomography algorithms say that accurate reconstruction requires infinitely many x-rays angles and infinitely many X-rays. In practice, this is impossible.

An important result was the Kotelnikov theorem [10], which says how many X-rays (scans) should be performed to obtain a reconstruction of satisfactory quality, of course for a finite number of scan angles and for a finite number of rays for a given angle. After the X-rays are performed, on the basis of the information obtained in this way, it is possible to proceed to the process of reconstructing the internal structure of the X-rayed object. Reconstructive algorithms are responsible for this process. These algorithms can be divided into two main groups: analytical algorithms [11,12,13] and algebraic algorithms [14,15,16].

In the case of an incomplete data set, we obtain information (projections) of much worse quality, mainly due to the range of scanning angles. Such projections do not satisfy the assumptions of Kotelnikov’s theorem, so there is no guarantee that it would be possible to obtain a reconstruction of satisfactory quality. The research carried out by the authors showed that analytical algorithms cannot solve this type of problem (in terms of an incomplete data set), while algebraic algorithms, with appropriate adaptation to the problem of an incomplete data set, allow for obtaining such a reconstruction.

## 3. Methodology

The ideas of analytical and algebraic algorithms arose long before the construction of the first computer tomograph. Thus, in 1937, Stefan Kaczmarz presented an iterative algorithm for solving a system of linear equations and proved its convergence [17] (assuming that a given square matrix system of linear equations has exactly one solution). This algorithm was rediscovered in the 1970s by Gordon, Bender and Herman as a useful tool for the problem of image reconstruction (after proving the convergence of Kaczmarz’s algorithm also in [15]). Figure 3 shows a graphic interpretation of Kaczmarz’s algorithm.

Before we discuss the presented algorithm, let us first define the way in which the system of linear equations to be solved arises. In algebraic algorithms, we initially assume that the examined object is inscribed in a square (which of course is always possible), and then we discretize this square by dividing it into $n\times n=n^{2}$ smaller congruent squares (pixels). It is also assumed that this discretization is so dense (the number *n* is large enough) that can be assumed the function *f* describing the density distribution of the examined object is constant for each pixel. Such an assumption results in the fact that the energy loss $p_{i}$ of the *i*-th ray is presented as the sum of energy losses along the path of this ray during its passage through the tested object. Energy is only lost on those pixels that are in the path of this ray. The energy loss on such a pixel is directly proportional to the change in length (knowing the size of the square in which the tested object is inscribed, the discretization density and the equation of the line along which a given radius runs, we can determine this length) the common part of a given radius and a given pixel and to the unknown energy absorption capacity of this pixel (constant value of the function *f* on this pixel). By carrying out a number of such X-rays, each of them is an equation, the left side of which is the sum of the above-mentioned energy losses, and the right side is the total loss. Value of the function *f* for each pixel is an unknown. As a result, we get a system of equations:(2)AX=B,
where A is the matrix of the $r\times n^{2}$ dimension (*r* is the number of X-rays), B is the matrix (vector) of the r×1 dimension of the energy loss of each ray, and X is the matrix (vector) of the $n^{2}\times 1$ dimension of unknowns.

Because matrix A has specific properties (it is a rectangular, sparse, asymmetric and large-dimension matrix), classical algorithms are not useful in this case. However, the Kaczmarz algorithm, or algorithms based on it, can be successfully used, e.g., the most famous ART algorithm, in which, after selecting any initial solution $x^{0}\in R^{n^{2}}$, subsequent approximations of the exact solution are determined by the formula:(3)$$x^{k+1}=x^{k}+\lambda_{k}\frac{p_{i}-\langle x^{k},a_{i}\rangle }{∥a^{i}{∥}^{2}}a^{i},$$
where $k\in N_{0}$ is the solution index, $\lambda_{k}$ is the relaxation coefficient (for $\lambda_{k}=\lambda =1$, we obtain the classic Kaczmarz algorithm), $p_{i}$ is the *i*-th projection (*i*-th element of vector B), $a^{i}$ is the *i*-th row of the matrix A, 〈·,·〉 means the classical dot product, and ∥·∥ is the norm of a vector. Trummer showed [18] that, for $0<\lambda_{k}=\lambda <2$, i.e., for the constant value of the λ parameter, or for $0<lim inf\lambda_{k}\leq lim sup\lambda_{k}<2$, i.e., for the variable value of the λ parameter, the ART algorithm is convergent.

In Ref. [4], the authors approached the issue of reconstruction the interior of the examined object in an innovative way. In this paper, authors significantly increase the efficiency of this process through a new approach to the previously introduced idea. It was assumed that detected objects were represented as rectangles. This approach has advantages and is a good mathematical model of the studied phenomenon. Since there are relatively few detectable undesirable objects in the coal seam, we can limit to a relatively small number of rectangles representing these objects. The adopted shape is appropriate, despite the fact that the actual objects may not be rectangles, and the difference between a rectangle and a “non-rectangle” is negligible (small-size stone overgrowth is not a significant loss of the output quality).

After X-rays, we obtain a system of linear equations (Equation 2) and in the classic approach, this system of equations should be solved using the relationship (Equation 3). However, this step requires a lot of computation and therefore a long time to reconstruction. Thus far, rectangles representing the density distribution have been detected by using heuristic algorithms. We use a similar approach in this paper. The innovation of this approach will be the representation of these rectangles. Previously, order of rectangles was important, which really does not matter. The representation of a single rectangle was also ambiguous (in means of the order of the vertices). In this paper, we eliminate these two inconveniences, thanks to which the time of finding the rectangles should be significantly shortened.

Such an approach is necessary for the possibility of using heuristic algorithms in a computed tomography task. We can find articles (see, e.g., [19,20,21]) in which the authors combine these two methods, but, in these approaches, heuristic algorithms are used to analyze the image, after the system of equations (Equation 2) has been solved. The lack of works in which heuristic algorithms were used was dictated by their feature—these algorithms work well with the optimization of functions with a relatively small number of arguments. In the standard approach, there would be as many arguments as there would be unknowns in the system of equations (Equation 2). Thus, it would be enough to perform a not very dense discretization (e.g., assuming n=20, and the variables would then be $n^{2}=400$), which would make it impossible to use the heuristic algorithm. The innovative approach is based on the fact that the unknowns (variables) are the features identifying the rectangles that represent the searched objects in the considered area. These features are the position of the rectangle and the value of the function *f* in the rectangle. Another advantage of this approach is the insensitivity of the method to discretization density. If we place *m* transmitters on one of the walls of the coal seam, and *m* sensors on the other, and if each transmitter sends a beam to each sensor, the system of equations will have $m^{2}$ of equations and $n^{2}$ of unknowns (increasing the discretization density forces an increase in the number of sensor sources). In this approach, the complexity of the task does not directly depend on the discretization density as much as in the classical approach. The heuristic algorithm arranges rectangles (along with information about the value of the function *f* in these rectangles) in the area in such a way as to minimize the value of the function:(4)$$F=∥B-B^{*}{∥}^{2}=\sum_{i=1}^{m^{2}}{\left(b_{i}-b_{i}^{*}\right)}^{2},$$
where F is a minimized function, B is a projection vector (right side of the system Equation 2), $B^{*}$ is a projection vector created on the basis of the constant matrix A and a given approximation of the solution X, *m* is the number of sources (and sensors), $b_{i}$ and $b_{i}^{*}$, $i=1,2,\ldots ,n^{2}$, are, respectively, *i*-th element of vectors B and $B^{*}$; $n^{2}$ is the number of pixels.

## 4. Heuristic Algorithms

This section presents the heuristic algorithms used in the calculations to minimize the objective function (Equation 4) and reconstruction the sought rectangles. Algorithms of this type are widely used in various types of engineering tasks [22,23,24,25,26,27,28,29]. In this paper, we present and compare the following algorithms: *Aquila Optimizer* (AO), *Firefly algorithm* (FA), *Whale Optimization Algorithm* (WOA), *Butterfly Optimization Algorithm* (BOA) and *Dynamic Butterfly Optimization Algorithm* (DBOA). In order to describe the methods, we introduce the following notations:N−populationsize,Dim−dimensionsize,T−numberofiterations,
$$x_{j}^{t}=\left[x_{j1}^{t},x_{j2}^{t},\ldots ,x_{jDim}^{t}\right]-j-\mathrm{th}\;\mathrm{solution}\;\mathrm{in}\;t-\mathrm{th}\;\mathrm{iteration},\;\;\;j=1,2,\ldots ,N,$$
Fit−costfunction.

### 4.1. Aquila Optimizer

Aquila is a type of bird from the hawk family (*Accipitridae*) found in Eurasia, North America, Africa and Australia. The way in which these birds hunt their prey was the inspiration for the creation of the algorithm of searching for a minimum of functions. Aquila predators base their hunting behavior on four main techniques:Expanded exploration. The predator soars high in the air with a vertical tilt—this method is used for hunting birds in flight, where the Aquila rises high above the ground. Upon examining the prey, Aquila goes into a long, low-angle slide with increasing speed. Aquila needs a height above its prey for this method to be effective. Just before the attack, the wings and tail are spread out and the claws bent forward to catch the prey. This process in the algorithm is described by the following equation:
(5)$$x^{t+1}=\left(1-\frac{t}{T}\right)\;x_{best}^{t}+\left(x_{m}^{t}-rd\;x_{best}^{t}\right),$$
where $x^{t+1}$ is solution in t+1-th iteration, $x_{best}^{t}$ is the best solution obtained in the *t*-th iteration (so far) and reflects the approximate location of the prey (which is the optimization goal). By $x_{m}^{t}$, we denote the mean solution in the iteration number *t* and calculate it as:
(6)$$x_{m}^{t}=\frac{1}{N}\sum_{i=1}^{N}x_{i}^{t}.$$The expression $1-\frac{t}{T}$ is responsible for the scope of the search. The closer the value of *t* is to the total iterations *T*, the more narrow the scope of the search. Value of rd is a random number from [0,1] interval.Narrowed exploration. In this method, when the aquila is high above the prey, it begins to circle around it and prepare to land and attack. This technique is called short stroke contour flight. Aquila narrowly explores the selected area where the prey is located and prepare to attack. This behavior is mathematically represented by the equation:
(7)$$x^{t+1}=Levy_{D}\;x_{best}^{t}+x_{r}^{t}+rd\;\left(r\cos\phi -r\sin\phi \right),$$
where $x^{t+1}$ is solution in iteration t+1, $x_{best}^{t}$ is the best solution obtained in the iteration *t*, $x_{r}^{t}$ is a random solution from iteration *t*, and rd is random number from range [0,1]. $Levy_{d}$ is value of the Levy flight distribution function. We compute the values of this function as follows:
(8)$$Levy_{D}=\frac{s\;u\;\sigma }{{\left|v\right|}^{\frac{1}{\beta }}},$$
where s,β are constants u,v are random numbers from interval [0,1], and σ is computed by formula [30]:
(9)$$\sigma =\frac{\Gamma \left(1+\beta \right)\sin\left(\frac{\pi \beta }{2}\right)}{\Gamma \left(\frac{1+\beta }{2}\right)2^{\frac{\beta -1}{2}}\beta }.$$The expression rcosϕ−rsinϕ is intended to simulate a spiral flight. Parameters *r* and ϕ are calculated as follows:
(10)$$r=r_{1}+V\;D_{1},\;\;\;\;\theta =-\xi \;D_{1}+\frac{3\pi }{2},$$
where $r_{1}$ is a fixed integer from {1,2,…,30}, V,ξ are small constants, $D_{1}$ is an integer from {1,2,…,Dim}.Expanded exploitation. In this step, the prey is already located and the aquila launches an initial vertical attack to see the victim’s reaction. Here, the aquila exploits a selected target area to get as close to its prey as possible. We transfer this behavior to the mathematical description of the method with the following equation:
(11)$$x^{t+1}=\alpha \;\left(x_{best}^{t}-x_{m}^{t}\right)-rd+\delta \;\left(rd\;(uBound-lBound)+lBound\right),$$
where $x^{t+1}$ is the solution in the next iteration after *t*, $x_{best}^{t}$ is the best solution obtained in the iterations so far, and $x_{m}^{t}$ is the population mean solution defined by the formula (Equation 6). As before, rd is a random number between [0,1], lBound, uBound define the problem domain (lower and upper bound), and α and δ are exploitation adjustment constants’ parameters.Narrowed exploitation. In this technique, the aquila has already approached the prey and attacks with stochastic movements, approaching and grabbing the victim. Mathematically, this process is described by the equation:
(12)$$x^{t+1}=QF\;x_{best}^{t}-\left(G_{1}\;rd\;x_{m}^{t}\right)-G_{2}\;Levy_{D}+r_{d}\;G_{1},$$
where $x^{t+1}$ is solution in t+1 iteration, and QF denotes the so-called quality function:
(13)$$QF=t^{\frac{2rd-1}{{\left(1-T\right)}^{2}}}.$$$G_{1}$ and $G_{2}$ are calculated by:
(14)$$G_{1}=2rd-1,\;\;\;G_{2}=2{\left(t-T\right)}^{2}.$$These parameters reflect the way the predator flies, and in the case of the algorithm, we can fine-tune the algorithm with it.

In nature, all these techniques are mixed in hunting prey, and in the case of the algorithm described here, the formulas (Equation 5)–(Equation 14) form the set of four transformations that make up the AO algorithm. Pseudocode Algorithm 1 shows steps of the AO algorithm. Details related to the Aquila Optimizer and application of it can be found in [30,31,32].
**Algorithm 1:** Aquila Optimize (AO) pseudocode.1:**Initialization part.**2:Setting up parameters of AO algorithm.3:Random initialization of starting population $\left\{x_{0}^{0},x_{1}^{0},\ldots ,x_{N}^{0}\right\}$.4:**Iterative part.**5:**for**iterationt=0,1,…,T−1**do**6:    Calculate the values of the cost function Fit for each element in the population.7:    Determine the best individual $x_{best}$ in the population.8:    **for** k=1,2,…,N **do**9:         Update mean individual $x_{m}$ in the population.10:        Update algorithm parameters based on population number and random11:        values ($G_{1},G_{2},QF$).12:        **if** $\mathrm{iteration}t\leq \frac{2}{3}T$ **then**13:           **if** rd<0.5 **then**14:               Do the step *expanded exploration* (Equation 5) updating solution $x_{k}^{t}$, obtaining15:               $x_{k}^{t+1}$16:               **if** $Fit\left(x_{k}^{t+1}\right)<Fit\left(x_{k}^{t}\right)$ **then** $x_{k}^{t}=x_{k}^{t+1}$17:               **end if**18:               **if** $Fit\left(x_{k}^{t+1}\right)<Fit\left(x_{best}^{t}\right)$ **then** $x_{best}^{t}=x_{k}^{t+1}$19:               **end if**20:           **else**21:               Do the step *narrowed exploration* (Equation 7) updating solution $x_{k}^{t}$.22:               Then, obtain $x_{k}^{t+1}$23:               **if** $Fit\left(x_{k}^{t+1}\right)<Fit\left(x_{k}^{t}\right)$ **then** $x_{k}^{t}=x_{k}^{t+1}$24:               **end if**25:               **if** $Fit\left(x_{k}^{t+1}\right)<Fit\left(x_{best}^{t}\right)$ **then** $x_{best}^{t}=x_{k}^{t+1}$26:               **end if**27:           **end if**28:        **else**29:           **if** rd<0.5 **then**30:               Do the step *Expanded exploitation* (Equation 11) updating solution $x_{k}^{t}$.31:               Then, obtain $x_{k}^{t+1}$32:               **if** $Fit\left(x_{k}^{t+1}\right)<Fit\left(x_{k}^{t}\right)$ **then** $x_{k}^{t}=x_{k}^{t+1}$33:               **end if**34:               **if** $Fit\left(x_{k}^{t+1}\right)<Fit\left(x_{best}^{t}\right)$ **then** $x_{best}^{t}=x_{k}^{t+1}$35:               **end if**36:           **else**37:               Do the step *narrowed exploitation* (Equation 12) updating solution $x_{k}^{t}$.38:               Then, obtain $x_{k}^{t+1}$39:               **if** $Fit\left(x_{k}^{t+1}\right)<Fit\left(x_{k}^{t}\right)$ **then** $x_{k}^{t}=x_{k}^{t+1}$40:               **end if**41:               **if** $Fit\left(x_{k}^{t+1}\right)<Fit\left(x_{best}^{t}\right)$ **then** $x_{best}^{t}=x_{k}^{t+1}$42:               **end if**43:           **end if**44:        **end if**45:    **end for**46:**end for**47:**return**$x_{best}$.

### 4.2. Firefly Algorithm

The firefly algorithm is a swarm metaheuristics developed in 2008 by Xin-She Yang [33]. It is inspired by the social behavior of fireflies (insects from the Lampyridae family) and the phenomenon of their bioluminescent communication. The algorithm is intended to constrained continuous optimization problems. It is dedicated to minimize cost function Fit, i.e., finding $x^{*}$ that:(15)$$Fit\left(x^{*}\right)=\min_{x\in S}Fit\left(x\right),$$
where $S\subset R^{n}$.

Now, we present the basics assumptions of the fireflies algorithm:Assume that there is a swarm of *N* agents (fireflies) solving the optimization problem iteratively, where $x_{j}^{t}$ represents the solution for the firefly *j* in the *k* iteration, and $Fit(x_{i}^{k})$ stands for its cost.Each firefly has a distinguishing β attractiveness, which determines how strongly it attracts other members of the swarm.As attractiveness of the firefly, a decreasing distance function should be used, e.g., as suggested by Yang, the following function:
(16)$$\beta =\beta_{0}e^{-\gamma r_{ij}^{2}},$$
where $\beta_{0}\in \left[0,1\right]$ and γ are the algorithm parameters, respectively: maximum attractiveness and absorption coefficient. Distance function is denoted by $r_{ij}=dist\left(x_{i},x_{j}\right)$. It is common that dist can be taken as $\left|\right|x_{i}-x_{j}{\left|\right|}_{2}$. Often, the absorption coefficient is $\gamma =\frac{\gamma_{0}}{r_{max}}$, where $\gamma_{0}\in \left[0,1\right]$, and $r_{max}=\max_{i,j}dist\left(x_{i},x_{j}\right)$.Each swarm member is characterized by the luminosity $l_{j}$, which can be directly expressed as the inverse of the value of the cost function $Fit(x_{j})$.Initially, all fireflies are placed in the search space *S* (randomly or using some deterministic strategy).To effectively explore the solution space, it is assumed that each firefly *j* changes its position iteratively, taking into account two factors: attractiveness of other members of the swarm with greater luminosity $l_{i}>l_{j}$, ∀i=1,…,N,i≠j, which decreases with distance, and a random step $u_{j}$.For the brightest firefly, only the above-mentioned random step is applied.The movement of the firefly *j* to the brighter firefly *i* is described by equation:
(17)$$x_{j}^{t+1}=x_{j}^{t}+\beta \left(x_{i}^{t}-x_{j}^{t}\right)+u_{j},$$
where the second term corresponds to the attraction, and the third term is a random number from interval $\left(\min_{u_{j}},\max_{u_{j}}\right)$.

The pseudocode Algorithm 2 describes in steps the fireflies algorithm. More about the firefly algorithm can be found in [33,34].
**Algorithm 2:** Firefly algorithm (FA) pseudocode.1:**Initialization part.**2:Setting up parameters of FA algorithm parameters $N,\beta_{0},\gamma ,\min u_{j},\max u_{j},\mathrm{for}\;j=1,\ldots ,Dim,\;\;Dim-\mathrm{dimension}\mathrm{size}$.3:Random initialization of starting population $\left\{x_{0}^{0},x_{1}^{0},\ldots ,x_{N}^{0}\right\}$.4:**Iterative part.**5:**for**iterationt=0,1,…,T−1**do**6:    Determine $x_{best}^{t}=\min_{x_{k}}Fit\left(x_{k}^{t}\right)$ and $k_{best}=\min_{k}Fit\left(x_{k}^{t}\right)$.7:    **for** k=1,2,…,N **do**8:        **if** $k\neq k_{best}$ **then**9:           **for** j=1,2,…,N **do**10:               **if** $Fit\left(x_{j}^{t}\right)<Fit\left(x_{k}^{t}\right)$ **then**11:                   Calculate: $r_{kj}$, $\beta =\beta_{0}e^{-\gamma r_{kj}^{2}}$, $u_{k}=rand\left(\min u_{k},\max u_{k}\right)$.12:                   Transform *k*-th solution $x_{k}^{t}$ according to the Formula (Equation 17).13:               **end if**14:           **end for**15:        **else**16:           Calculate $u_{k_{best}}=rand\left(\min u_{k_{best}},\max u_{k_{best}}\right)$17:           Convert the best solution $x_{best}^{t}$ based on the formula:
$$x_{best}^{t+1}=x_{best}^{t}+u_{k_{best}}.$$18:        **end if**19:    **end for**20:**end for**21:**return**$x_{best}$.

### 4.3. Whale Optimization Algorithm

Whale Optimization Algorithm (WOA) discussed in this paragraph is inspired by the hunting behavior of whales. These huge mammals hunt their prey in a specific way, which is called the bubble-net feeding method. It has been observed that this foraging takes place by the formation of characteristic bubbles along a circle or spiral shaped path. Whale-watching scientists noticed two maneuvers related to the bubble and called them “up spirals” and “double loops”. In the first method, predators dive approximately 12 m downwards and then begin to form a spiral-shaped bubble around their prey and swim towards the surface. The second hunting technique involves two different steps: coral loop and capture loop. It is worth mentioning here that the bubble-net feeding method is a unique behavior that can only be observed in whales. In this paper, the whale hunting technique serves as an inspiration for the minimum cost function search algorithm. Below, we present the basic principles of the Whale Optimization Algorithm.

Since the position of the optimal solution in the search space is unknown, the WOA algorithm assumes that the target victim is currently the best candidate for a solution or is close to the optimum. Once the best search agent is defined, the remaining agents try to update their positions towards the best search agent. This behavior is represented by the following equations:
(18)$$r=|c\cdot x_{best}^{t}-x^{t}|,$$
(19)$$x^{t+1}=x_{best}^{t}-a\cdot r,$$
where $x^{t}$ is a certain solution in the *t*-th iteration, $x_{best}^{t}$ is the best solution in iteration *t*, a,c are vectors of coefficients, and operation · is defined:
$$c\cdot x^{t}=\left[c_{1}x_{1}^{t},\ldots ,c_{Dim}x_{Dim}^{t}\right],\;\;Dim-\mathrm{dimension}\mathrm{size}.$$Vectors a,c are calculated by formula:
(20)a=2v·rd−v,c=2rd,
where v is the Dim length vector of numbers which decrease from iteration to iteration. In the case under consideration, we assume $v_{i}=2\frac{T-t}{T}$ for all i=1,2,…,Dim, where *T* is total number of iterations. rd denotes the vector of length Dim random numbers from interval [0,1].The mathematical model of the bubble-net behavior is based on the following two assumptions:−shrinking encircling mechanism is reducing the value of the vector v from iteration to iteration. As a result, the value range of the vector a in Equation (Equation 20) also becomes smaller;−spiral updating position describes a relation between the whale’s and the prey’s position to mimic the spiral-shaped movement of the whale. We describe this process mathematically with the following equation:
(21)$$x^{t+1}=\left|x_{best}^{t}-x^{t}\right|e^{bl}\cos\left(2\pi l\right)+x_{best}^{t},$$
where constant *b* and random number l∈[−1,1] define the shape of the spiral movement.The whales move around the prey in a contracting circle and at the same time along a spiral path. To model this behavior, we assume that there is a 50% probability to choose between a contracting, orbiting mechanism or a spiral model to update the position of the whales during optimization. The mathematical model is described by the following equations:
(22)$$x^{t+1}=x_{best}^{t}-a\cdot r,\;\;\;\mathrm{for}p<0.5,$$
(23)$$x^{t+1}=\left|x_{best}^{t}-x^{t}\right|e^{bl}\cos\left(2\pi l\right)+x_{best}^{t},\;\;\;\mathrm{for}p\geq 0.5,$$
where *p* is a random number from range [0,1].A similar approach was applied, based on the range variability of the vector a, for the exploration phase. Hence, we use a with random values greater than 1 or less than −1 to force the agent to move away from the best position. In practice, we obtain $|a_{i}|>1$, for all i=1,2,…,Dim. Unlike the exploitation phase, here we update the agent position according to a randomly chosen solution instead of the best one found so far. Since $|a_{i}|>1$, this mechanism places emphasis on exploration and allows the WOA to perform a global search. This process mathematically can be described with the following equation:
(24)$$r=|x\cdot x_{rand}^{t}-x^{t}|,$$
(25)$$x^{t+1}=x_{rand}-a\cdot r,$$
where $x_{rand}$ is a random agent.

More about the algorithm and its applications can be found in the papers [35,36]. Based on the above rules, we now present the pseudocode Algorithm 3 of the WOA.
**Algorithm 3:** Whale optimization algorithm (WOA) pseudocode.1:**Initialization part.**2:Setting up parameters of WOA.3:Random initialization of starting population $\left\{x_{0}^{0},x_{1}^{0},\ldots ,x_{N}^{0}\right\}$.4:Calculating the value of the cost function Fit for each individual $x_{i}^{0}\;\left(i=1,2,\ldots ,N\right)$ in the population.5:Determining the best agent in the population $x_{best}^{0}$.6:**Iterative part.**7:**for**iterationt=0,1,…,T−1**do**8:    **for** k=1,2,…,N **do**9:         Determine values of a,c,l,p.10:        **if** p<0.5 **then**11:           **if** |a|<1 **then**12:               Determine a new solution according to the Formula (Equation 19).13:           **end if**14:           **if** |a|≥1 **then**15:               Determine a random solution $x_{rand}$.16:               Determine a new solution according to the Formula (Equation 25).17:           **end if**18:        **end if**19:        **if** p≥0.5 **then**20:           Determine a new solution according to the Formula (Equation 23).21:        **end if**22:    **end for**23:    Calculating the value of the cost function Fit for each individual $x_{i}^{t+1}\;\left(i=1,2,\ldots ,N\right)$ in the population.24:    Determine the best agent in the population $x_{best}^{t+1}$.25:**end for**26:**return**$x_{best}$.

### 4.4. Butterfly Optimization Algorithm

Butterflies use their sense of smell, taste and touch to find food and partners. These senses are also helpful in migrating from one place to another, escaping from a predator, and laying eggs in the right places. Among senses, the smell is the most important one that helps the butterfly find food, usually nectar—even if it is far away. To find a source of nectar, butterflies use sense receptors scattered throughout the butterfly’s body (e.g., on foreheads, legs). These receptors are in fact nerve cells on the surface of the butterfly’s body and are called chemoreceptors.

Based on scientific observations, it was found that butterflies can sense, and therefore locate, the source of an odor very accurately. In addition, they can distinguish smells as well as sense their intensity. These abilities and behavior of butterflies were used to develop the Butterfly optimization algorithm (BOA) [37]. The butterfly produces a scent of a certain intensity, which is related to its ability to move from one location to another. The fragrance is sprayed from a distance and other butterflies can sense it and thus communicate with each other, creating a collective knowledge network. A butterfly’s ability to sense the scent of another butterfly became the basis for the development of algorithm to search a global optimum. In another scenario, when the butterfly cannot sense the smell of its surroundings, it will move randomly. In the proposed algorithm, this phase is referred to as local search.

In order to understand how fragrance is computed in the BOA, one first needs to understand how modality, smell, sound, light, temperature, etc. are processed by the stimulus. The whole concept of sensing and modality processing is based on three parameters: sensory modality (*c*), stimulus intensity (*I*) and power exponent (*a*). By modality, we can understand smell, sound, light or temperature. In the case of BOA, the modality is fragrance. *I* is the stimulus intensity. In the BOA, *I* is correlated with butterfly fitness. This means that, when a butterfly emits more smell, the remaining butterflies in that environment can sense it and become attracted to a highly scented butterfly. Fragrance *f* is directly proportional to the power of intensity *I* with the exponent *a*:(26)$$f=cI^{a},$$
where *f* is fragrance, *c* is the sensory modality, *I* is the stimulus intensity, and *a* is the power exponent depending on the modality. In most cases, in implementation, we can take *a* and *c* from the [0,1] interval. The *a* parameter is a modality dependent power exponent, which means that it has absorption variability. Thus, the *a* parameter controls the behavior of the algorithm. Another important parameter is *c*, which is also a key parameter in determining BOA behavior. Theoretically, c∈[0,∞), but, in practice, it is in the [0,1] range. The *a* and *c* values have a decisive influence on the convergence speed of the algorithm. The selection of these parameters is important and depends on the characteristics of the problem under consideration.

The BOA algorithm is based on the following rules:Each of the butterflies gives off a fragrance with a different intensity. Thanks to the smell, butterflies can communicate.The movement of the butterfly occurs in two ways: towards an individual emitting a stronger smell, or at random.Global search move is represented by the formula:
(27)$$x^{t+1}=x^{t}+\left(r^{2}\;x_{best}^{t}-x^{t}\right)f,$$
where $x^{t}$ denotes the location of butterfly (agent) in the *t*-th iteration, $x^{t+1}$ is the transform location of butterfly in the t+1-th iteration t+1, $x_{best}$ is the position of the best butterfly in the *t*-th iteration, *f* is a fragrance of $x^{t}$, and *r* is a random number from [0,1].Local search move is described by the following equation:
(28)$$x^{t+1}=x^{t}+\left(r^{2}\;x_{r1}^{t}-x_{r2}^{t}\right)f,$$
where $x_{r1}^{t}$ i $x_{r2}^{t}$ are randomly chosen agents from iteration *t*.

Information about applications of BOA can be found in [38,39]. Scheme Algorithm 4 presents a pseudocode of BOA.
**Algorithm 4:** Butterfly optimization algorithm (BOA) pseudocode.1:**Initialization part.**2:Setting up parameters of BOA algorithm. *N*, Dim, *c*, *a* and *p*.3:Random initialization of starting population $\left\{x_{0}^{0},x_{1}^{0},\ldots ,x_{N}^{0}\right\}$.4:Calculating the value of the cost function Fit (thus the intensity of the stimulus I=Fit) for every agent $x_{i}^{0}\;\left(i=1,2,\ldots ,N\right)$ in population.5:**Iterative part.**6:**for**iterationt=0,1,…,T−1**do**7:    **for** k=1,2,…,N **do**8:        Calculate value of fragrance for $x_{k}^{t}$ using Formula (Equation 26).9:     **end for**10:    Determine the best agent $x_{best}^{t}$ in the population.11:    **for** k=1,2,…,N **do**12:        Determine a random number *r* from interval [0,1].13:        **if** r<p **then**14:           Transform solution $x_{k}^{t}$ according to the Formula (Equation 27).15:        **else**16:           Transform solution $x_{k}^{t}$ according to the Formula (Equation 28).17:        **end if**18:    **end for**19:    Change the parameter value *a*.20:**end for**21:**return**$x_{best}$.

### 4.5. Dynamic Butterfly Optimization Algorithm

Now, we present an improved version of the BOA called in the literature Dynamic Butterfly Optimization Algorithm (DBOA) [40]. This improvement consists of adding a novel local search algorithm based on mutation operator (LSAM) at the end of the BOA main loop. LSAM transforms the population (best solution first) using the mutation operator. If the new solution (obtained after mutation) turns out to be better than the previous one (before mutation), then it replaces the previous one. This will transform the agents population. The LSAM algorithm is presented in the scheme Algorithm 5. In the literature, there are many papers about the applications of the butterfly algorithm and its modifications (see [41,42,43]).

The mutation operator occurring in the schema Algorithm 5 transforms each coordinate of the solution $x=[x^{1},x^{2},\ldots ,x^{Dim}]$ substituting it with a random number from the normal distribution, as shown in the formula below:(29)$$x_{new}^{i}∼N\left(x^{i},\sigma \right),$$
where $x^{i}$ is mean and σ=0.1(uBound−lBound) is standard deviation. Pseudocode of DBOA is shown in the scheme Algorithm 6.
**Algorithm 5:** Novel local search algorithm based on mutation operator (LSAM) pseudocode.1:$x_{best}$ – current best solution from BOA.2:$Fit_{best}=Fit\left(x_{best}\right)$ – value of objective function for current best solution.3:*I* – number of iterations, mr – mutation rate.4:**Iterative part.**5:**for**iterationi=0,1,…,I−1**do**6:    Calculate: $\;\;x_{new}=$  Mutate($x_{best},mr$), $\;\;\;Fit_{new}=Fit\left(x_{new}\right)$.7:    **if** $Fit_{new}<Fit_{best}$ **then**8:        $x_{best}=x_{new}$, $\;\;Fit_{best}=Fit_{new}$.9:    **else**10:        Determine a random solution $x_{rnd}$ from the population, different from $x_{best}$.11:        Calculate the value of the objective function $Fit_{rnd}=Fit\left(x_{rnd}\right)$.12:        **if** $Fit_{new}<Fit_{rnd}$ **then**13:           $x_{rnd}=x_{new}$14:        **end if**15:    **end if**16:**end for**

**Algorithm 6:** Dynamic butterfly optimization algorithm (DBOA) pseudocode.
1:
**Initialization part.**
2:Setting up parameters of BOA algorithm. *N* – population size, Dim – problem dimension, *c* – sensor modality, parameters *a* and *p*.3:Random initialization of starting population $\left\{x_{0}^{0},x_{1}^{0},\ldots ,x_{N}^{0}\right\}$.4:Calculating the value of the cost function Fit (thus the intensity of the stimulus I=Fit) for every agent $x_{i}^{0}\;\left(i=1,2,\ldots ,N\right)$ in population.5:
**Iterative part.**
6:
**for**

iterationt=0,1,…,T−1

**do**
7:    **for** k=1,2,…,N **do**8:        Calculate value of fragrance for $x_{k}^{t}$ using Formula (Equation 26).9:     **end for**10:    Determine the best agent $x_{best}^{t}$ in the population.11:    **for** k=1,2,…,N **do**12:        Determine a random number *r* from interval [0,1].13:        **if** r<p **then**14:           Transform solution $x_{k}^{t}$ according to the Formula (Equation 27).15:        **else**16:           Transform solution $x_{k}^{t}$ according to the Formula (Equation 28).17:        **end if**18:    **end for**19:    Change the parameter value *a*.20:    Apply the LSAM algorithm to transform the agents population.21:
**end for**
22:**return**$x_{best}$.


## 5. Results

Selected algorithms are quite commonly used in various types of optimization problems, as evidenced by a large number of scientific publications. A lot of numerical experiments and research have shown that these algorithms can be adapted to the requirements of the considered problem. Numerical experiments were performed for all the heuristic algorithms presented in the previous section. All algorithms were implemented by the authors and then tested on the following functions: Bent Cigar Function, Rosenbrock Function, Rastrigin Function and HGBat Function.

Fine-tuning of the algorithms, based on the literature, own experiences and experiments on test functions, consisted of determining the range of values for each parameter. From this range, 20 values were uniformly selected and tested on numerous examples relating to the detection of a single object. Numerical experiments were performed for testing all possible combinations of selected parameters multiple times with the same data set. As a result, the best sets of parameters were selected for each of the algorithms. Further calculations were performed for them.

The essential step of the research is a series of numerical experiments for tomography tasks. First, 50 tests were prepared for which the problem of tomography comes down to the detection of one object. These tasks were primarily used to calibrate the parameters of individual algorithms. At this stage, it was found that the AO algorithm is not suitable for this type of issue. Numerous attempts to select algorithm parameters led to unsatisfactory results, and the best approximate solution of the problem finally obtained was burdened with an unacceptable error of 2%. The second stage of the experiments was to find a solution in the form of two disjoint anomalies. In this case, a random set of test tasks was also generated. Each of them consisted of identifying two separate anomalies in the form of rectangles. In this case, all algorithms were called for the parameters (except for the size of the population and the number of iterations) calibrated in the first stage of the research. For this significantly more complex task, where the functional depends on 10 parameters, the size of the population and the maximum number of iterations were selected by means of a numerical experiment. Tasks of this type turned out to be significantly more difficult for WOA and BOA algorithms. This stage turned out to be too difficult for these algorithms and, in an acceptable time, these algorithms failed to find a satisfactory approximate solution.

Figure 4 presents a comparison of the dependence of the functional value on the number of calls to the objective function for the examples with one and two detected objects. The left figure shows that the value of the functional for the AO algorithm is too large (not at all zero). In the case of tasks with two searched objects, the WOA and BOA algorithms are not able to deal with, which is illustrated by the graphs in the figure on the right.

The last third stage of the research was to identify three disjoint anomalies. At this stage, the experiments were carried out for two algorithms: FA and DBOA because only these algorithms reconstruct two anomalies in a satisfactory time. In this case, similar to previous cases, experiments were also carried out for 50 different test tasks. For these tasks, the minimized functional depended on 15 arguments. The threefold increase in the number of parameters in relation to the first task significantly influenced the extension of the algorithm’s operation time, which was associated with the need to multiply the population size and the number of iterations.

Now, we present a comparison of the obtained results for the two tested algorithms at this stage of the experiments. Figure 5 presents comparisons of the exact solution (areas bordered with a solid line) with the approximate solution (areas bordered with a dashed line) obtained with the FA algorithm for selected iterations. The height of the areas (they are cuboids) has been marked according to the color scale. The presented results show that the FA algorithm, after a relatively short number of iterations (it= 20), “approaches” the exact solution, and obtaining a satisfactory result requires the performance of another 80 iterations.

Similarly, Figure 6 presents a summary of results in selected iterations for the second tested algorithm DBOA in the third stage of numerical experiments.

Comparing the obtained results, we can see that the convergence of both algorithms significantly decreases with the increase of successive iterations. In order to compare the effectiveness of the DBOA algorithm and the FA algorithm, Figure 7 compares the values of the objective function depending on the number of iterations (left figure) and the comparison of the objective function values depending on the number of calls the objective function (right figure). Comparing the number of iterations performed would indicate that the FA algorithm requires 10 times less iteration, but such a comparison does not indicate into the algorithm execution time, which in the case of the analyzed algorithms depends primarily on the number of calls to the objective function. As we can see, this number in the case of the DBOA algorithm is significantly smaller than in the case of the FA algorithm. In order not to blur the readability, the figure has been drawn in the range $\left[0,10^{6}\right]$ of calls to the objective function. It should be mentioned that the FA algorithm obtained a values of objective function close to zero only for $5\cdot 10^{6}$ calls to the objective function. Such comparisons show that the DBOA algorithm is definitely more effective in determining anomalies.

Table 1 presents the values of the objective function together with the number of calls to the objective function depending on the number of detected objects, which translates into the number of unknowns. The inscription “no results” means that the algorithm parameters were not selected so as to obtain repeatable results for many samples. The presented data clearly show that the DBOA algorithm turned out to be the best algorithm in the optimization problem presented in the paper.

## 6. Conclusions

The conducted research has shown that not all of the analyzed algorithms are suitable for the task formulated in this paper. The identifying of a larger number of anomalies is a complex process and the correct selection of the optimization method plays a crucial and important role. The experiments showed significant difficulties with the selection of appropriate parameters for the heuristic algorithms under study. The presented research results show that the DBOA algorithm handles the problem under consideration best. Analyzing the plot of the dependence of the value of the objective function on the number of its calls, it can be noticed that these algorithms quite quickly obtain solutions similar to the exact ones but require many calculations to obtain results close enough to the exact solution. Therefore, the DBOA method could be an initial method in a hybrid solution combining a heuristic algorithm for determining the starting point for classical computed tomography methods. The authors of this paper have future plans for research on the improvement and development of existing methods, as previously mentioned by creating a hybrid algorithm in which the DBOA algorithm would perform an exploratory function and the exploitation phase would be performed by classical computed tomography. The minimized functional turned out to be too difficult for some of the analyzed algorithms. It should be emphasized that this function, due to the qualitative nature of the detected anomalies (description by rectangles), should be generalized in further works by introducing a more complex geometry of the searched objects. Another point of research is therefore to change the geometry of the detected objects from rectangles to convex polygons. With this in mind, an important point of this stage of the research was to find an algorithm that allows for effectively detecting anomalies in the studied areas. It can be assumed that, for more geometrically complex objects, this task will be even more computationally difficult.

## Figures and Tables

**Figure 1 sensors-22-07297-f001:**
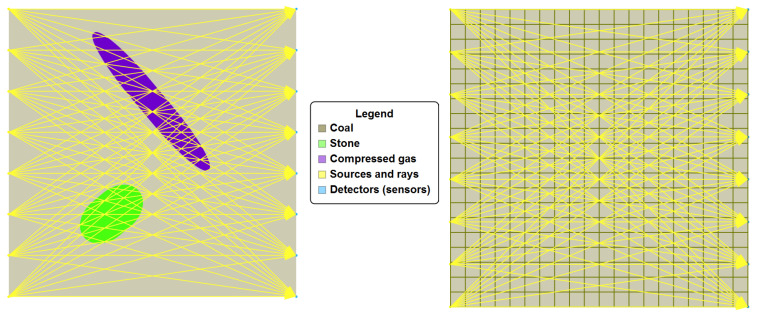
Graphical interpretation of obtaining the projection vector B and the coefficient matrix A of system (Equation 2).

**Figure 2 sensors-22-07297-f002:**
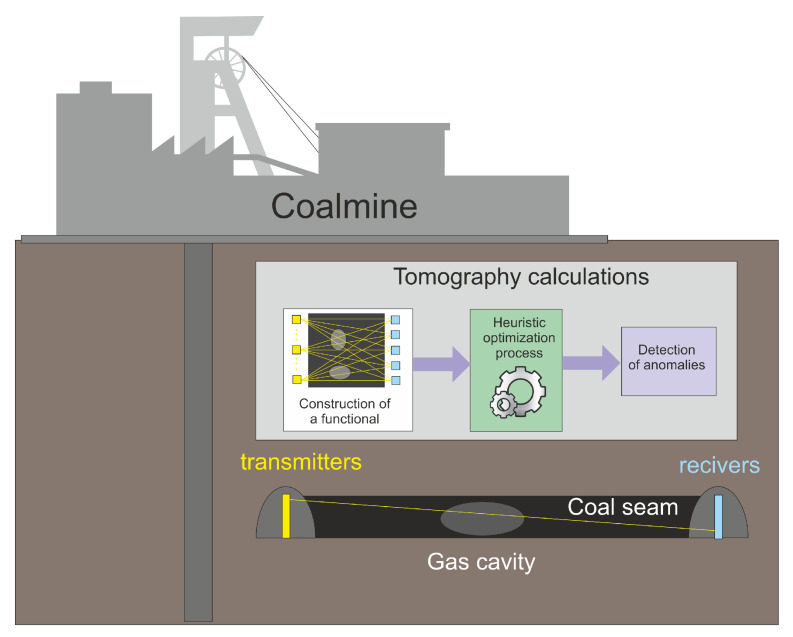
Illustrative drawing visualizing the operation of the anomaly detection system in coal seams.

**Figure 3 sensors-22-07297-f003:**
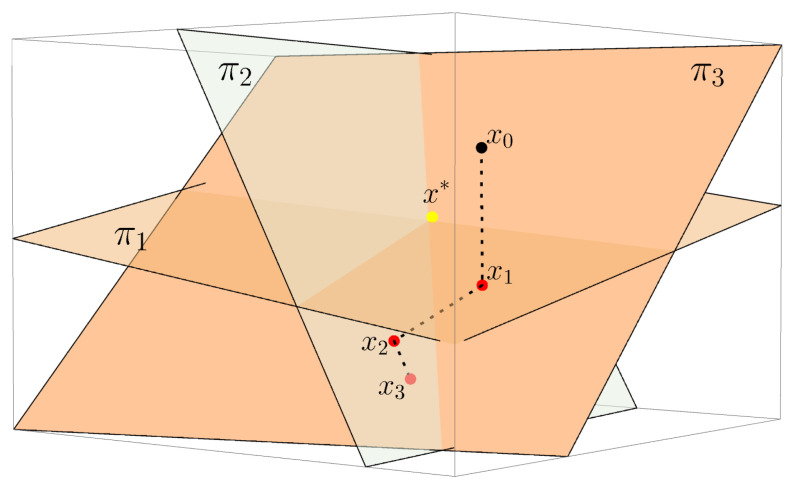
Geometric interpretation of the Kaczmarz’s algorithm.

**Figure 4 sensors-22-07297-f004:**
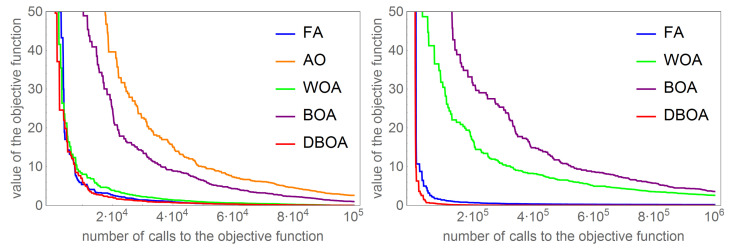
Comparison of the value of the objective function depending on the number of calls to the objective function for detecting one object (**left** figure) and detecting two objects (**right** figure).

**Figure 5 sensors-22-07297-f005:**
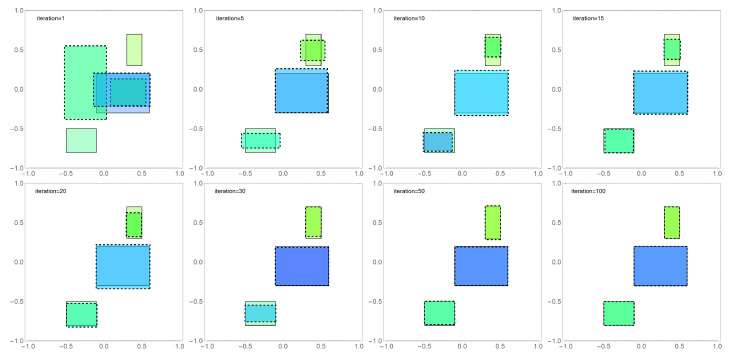
Graphical presentation of the results obtained with the FA algorithm for the population n=200 in selected iterations, where the reconstructed areas are outlined with a dashed line while the exact solution is outlined with solid line, and the area height is marked by color.

**Figure 6 sensors-22-07297-f006:**
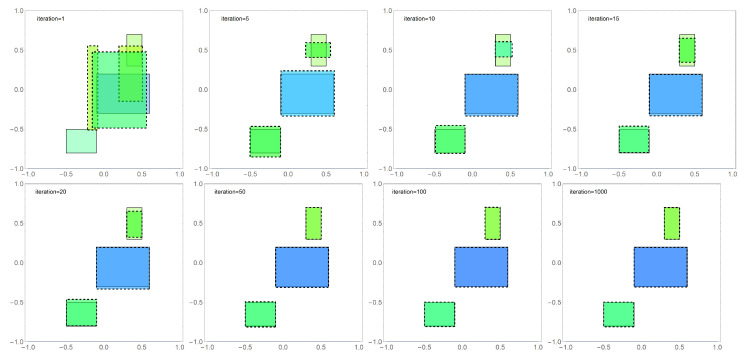
Graphical presentation of the results obtained with the DBOA algorithm for the population n=800 in selected iterations, where the reconstructed areas are outlined with a dashed line while the exact solution is outlined with solid line, and the area height is marked by color.

**Figure 7 sensors-22-07297-f007:**
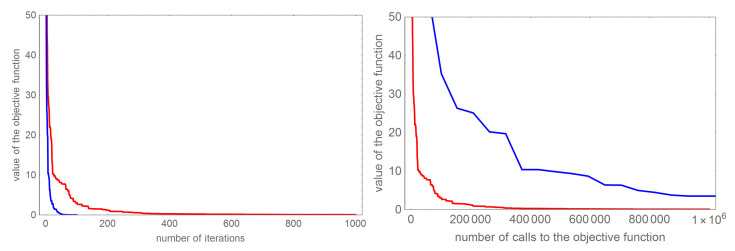
The **left** figure shows a comparison of the dependence of the objective function value on the number of iterations, while the **right** one shows the dependence of the objective function value on number of calls to the objective function. The results obtained with the DBOA algorithm are marked in red, and the results in blue are marked for the FA algorithm.

**Table 1 sensors-22-07297-t001:** Comparison of the number of calls to the objective function ($n_{f}$) and the value of the F function depending on the number of detected objects for the analyzed algorithms.

Number of Detected Objects		1 (5 Variables)		2 (10 Variables)		3 (15 Variables)	
		$n_{f}$	F	$n_{f}$	F	$n_{f}$	F
algorithm	AO	$1.23\cdot 10^{6}$	0.62	no results		no results	
	FA	$3.33\cdot 10^{5}$	$1.01\cdot 10^{-11}$	$3.88\cdot 10^{6}$	$8.56\cdot 10^{-6}$	$5.19\cdot 10^{6}$	$1.01\cdot 10^{-3}$
	WOA	$4.33\cdot 10^{5}$	$4.82\cdot 10^{-9}$	$4.23\cdot 10^{7}$	0.81	no results	
	BOA	$8.77\cdot 10^{5}$	$2.29\cdot 10^{-10}$	$6.31\cdot 10^{7}$	0.32	no results	
	DBOA	$2.21\cdot 10^{5}$	$5.02\cdot 10^{-10}$	$5.02\cdot 10^{5}$	$1.63\cdot 10^{-6}$	$1.00\cdot 10^{6}$	$8.01\cdot 10^{-4}$

## Data Availability

Not applicable.

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
