# Peer review of "Application of Heuristic Algorithms in the Tomography Problem for Pre-Mining Anomaly Detection in Coal Seams"

_sensors, 2022, doi:10.3390/s22197297_

Round 1
Reviewer 1 Report
In the paper is presented a novel metaheuristic algorithm named Dynamic butterfly optimization algorithm (DBOA), which is formulated to deal with the tomography problem for pre-mining anomaly detection in coal seams. However, major revisions are required before it could be processed for possible publication.
1. In the abstract, Summarize the results and add the summary here in two/three sentences. Not acceptable to write general information like that.
2. The contributions of this work is not clear. I suggest author to specify the contributions clearly in introduction section.
3. The convergence behavior analysis is recommended in terms of function evaluation. The number of function evaluations of the proposed algorithm is visualized and compared with the other contenders.
4. Justify, how DBOA improves the solution quality compared to other standard algorithms like butterfly optimization, whale optimization and firefly algorithm.
5. Please improve your conclusion section by using some suggestions regarding the scope of future research.
Reviewer 2 Report
The manuscript presents research on a specific approach to the issue of computed tomography with an incomplete data set, and apply the approach to the tomography problem of coal seam anomaly detection before mining. Authors apply the approach in a specific field, which has good research significance. The citations in the article are standardized, and the article is expressed fluently,However, the article has the following problems:
(1) The purpose of this manuscript is to solve the actual problem of coal seam detection, but it makes assumptions about the objects detected in the coal seam. Please give a description of the consistency between the actual situation and the hypothetical situation.
(2) The logic of the manuscript is confusing, the research objectives should be introduced in the introduction, but authors separately introduce the research objectives in Section 4. There are many sections in the manuscript, and some sections can be combined. For example, the content of Section 4 can be introduced in the introduction, and Section 3 and Section 5 can be combined. Section 6 is a large introduction to existing heuristics algorithms; and very little about themselves work.
(3) Section 6 overly introduces five heuristic algorithms used in computing, and less introduces the innovative points of this manuscript. It is suggested that the article focuses on presenting the work that we have done and use more graphs to present it.
(4) A variety of algorithms are introduced in this manuscript, but only the comparison results of the two algorithms are analyzed in FIG. 6. If conditions permit, more algorithms can be compared. The x in "x-raying", "x-rayed" and "x-rays" needs to be capitalized, please check for similar problems in the full manuscript. The matrix symbols A, vector symbols B need to be bolded in formula (2) and (4). In the scientific and technical literature, vector and matrix symbols need to be bolded, please check for similar problems in the full manuscript. The first formula in Section 6.2 is not labeled, and the symbol S in the formula is not consistent with the symbol s stated in the manuscript.
(5) The manuscript does not give detailed indicators to evaluate the performance of the model.
For the above reasons, this paper is not recommended for publication in this journal.
Round 2
Reviewer 1 Report
All my comments are revised appropriately. I suggest the paper can be accepted for publication.
Reviewer 2 Report
The author has supplemented the manuscript and gave pertinent responses to some of the review comments. However, the manuscript also requires detailed responses to the following questions before publication:
1. The manuscript uses extensive text to introduce 5 heuristic algorithms. Please explain why these 5 heuristic algorithms are selected and no other heuristic algorithms are selected. What is the reason for these 5 heuristic algorithms being selected?
2. In addition to the simple comparison and application of the selected algorithm, can you add some improvements and innovations to the algorithm? Make the function of the algorithm more perfect.
